# MULTIREACT: MULTIMODAL TOOLS AUGMENTED REASONING-ACTING TRACES FOR EMBODIED AGENT PLANNING

## ABSTRACT

In the field of embodied AI, Large Language Models (LLMs) have demonstrated impressive proficiency in tasks involving simple reasoning. However, they face significant challenges when confronted with longer-horizon tasks described in abstract instructions, particularly those involving intricate visual concepts. These challenges stem from two main limitations. Firstly, LLMs, which heavily rely on text, struggle to cope with the demands of complex embodied tasks that require nuanced multimodal reasoning. Secondly, LLMs encounter difficulties in recognizing and autonomously recovering from intermediate execution failures. To overcome these limitations and enhance the planning capabilities of LLMs in embodied scenarios, we propose a novel approach called MultiReAct. Our framework made the following efforts: We utilize a parameter-efficient adaptation of a pre-trained visual language model, enabling it to tackle embodied planning tasks by converting visual demonstrations into sequences of actionable language commands. By leveraging CLIP as a reward model, we identify instances of sub-instruction execution failure, significantly increasing the success rate in achieving final objectives. We introduce an adaptable paradigm for embodied planning through in-context learning from demonstration, independent of the specific Visual Language Model (VLM), and low-level actor. Our framework supports two distinct low-level actors: an imitation learning agent and a code generation-based actor. Using the MultiReAct framework, we apply it to a diverse set of long-horizon planning tasks and demonstrate superior performance compared to previous LLM-based methods. The extensive experimental results underscore the effectiveness of our approach in addressing long-horizon embodied planning

## 1 INTRODUCTION

Large language models (LLMs) (Brown et al., 2020; OpenAI, 2023) have been utilized in high-level robot planning(Vemprala et al., 2023; Liang et al., 2022; Ahn et al., 2022), enabling robots to understand and interpret natural language instructions provided by human operators. Leveraging their impressive ability to generate language, LLMs act as a bridge by translating text-based directives into actionable commands, empowering robots to perform complex tasks. However, while LLMs have shown effectiveness in certain robot planning scenarios, their capability to handle abstract instructions and intricate visual descriptions remains limited. Challenges arise when faced with tasks that involve multifaceted, multi-step procedures or intricate operations, as LLMs struggle to generate precise and coherent action plans in response to such detailed instructions. The intricacies associated with such directives often exceed the capabilities of conventional LLM-based approaches, resulting in suboptimal performance and limited practical utility within long-horizon embodied planning.

Previous approaches (Liang et al., 2022; Vemprala et al., 2023) have utilized Large Language Models (LLMs) to execute predefined human-generated prompts, relying solely on their internal representations to formulate plans. However, these approaches lack grounding between language to the physical world, limiting their capacity for reactive reasoning and knowledge updates. Consequently, such methods are prone to issues like hallucination and error propagation throughout the reasoning process. Moreover, LLM-based techniques exhibit limitations in recognizing objects that are more complex or have unseen attributes, often resulting in hallucinations when encountering unfamiliar

object traits. To mitigate hallucinations and enhance alignment with real-world environments, there is a growing interest in introducing multi-modal reasoning. Huang et al. (2022); Zeng et al. (2022); Ahn et al. (2022) have adopted a multi-step approach that involves grounding detected objects in the scene to textual descriptions, employing language models to generate domain-specific plans, and using imitation learning-based controllers for plan execution. However, these approaches primarily employ visual language models (VLMs) for object detection and do not actively contribute to the decomposition of intricate instructions into actionable primitives suitable for execution by LLM agents. Addressing the challenge of handling complex, long-horizon instructions has remained a key limitation in prior multi-modal LLM-based methodologies.

Another significant challenge lies in addressing unexpected failures during the execution of sub-instructions. Text LLM alone methods often treat each action they suggest as inherently successful. Consequently, when an agent engages in long-horizon planning, failures in intermediate reasoning and action can lead to error propagation, ultimately compromising the accuracy of the final outcome.

In response to the challenges faced by LLM-based agents when confronted with complex, long-horizon instructions, we present MultiReAct—an embodied intelligence framework that seamlessly integrates multimodal tools into the reasoning and action processes to tackle a wide array of robotics tasks. To bolster the planning capabilities of the LLM-Agent, we introduce two pivotal enhancements: **(1)** Video Captioning: We incorporate a video captioning model to ground demonstrations in a series of language sub-instructions. This approach streamlines multimodal reasoning within the LLM while leveraging a Visual Language Model (VLM) as an auxiliary module to facilitate inference. **(2)** Reward Model: By adopting a pre-trained reward model, we furnish the LLM agent with contextual cues regarding the successful execution of sub-instructions. This reward mechanism serves as a guiding influence on the agent's subsequent actions within the given context.

The augmentation of multimodal tools possesses the following core advantages:

1. **Enhance alignment with visual demonstrations**: The generated plans exhibit a strong alignment with visual demonstrations. Leveraging Visual Language Models (VLMs), our approach transform abstract instructions into a sequence of sub-instructions containing actionable primitives. This alignment enhances the overall robustness and effectiveness of planning, demonstrating a high degree of excitability.

2. **Adaptation to a broad spectrum of uncommon objects**: Multimodal tools, generalized at the object level, facilitate seamless adaptation to an extensive range of uncommon objects. This inherent flexibility enhances the versatility of our approach, enabling it to accommodate objects with diverse characteristics.

3. **Robust detection of execution failures**: Multimodal tools excel in detecting failed executions with exceptional precision. This heightened error detection capability streamlines error correction processes, resulting in greater efficiency and reliability.

To summarize, our key contributions are the following: 1. The MultiReAct Framework: We introduce the MultiReAct framework, which integrates multimodal tools into the reasoning-acting traces generated by LLMs. This integration fosters in-context learning from demonstrations and significantly enhances the multimodal reasoning capabilities of LLMs. 2. CLIP-Based Reward Model: We propose a novel yet simple approach that employs CLIP as the reward model in the ReAct traces for detecting failed executions in long-horizon planning. Empirical experiments confirm the effectiveness of this approach in mitigating error propagation, ultimately elevating the success rate of long-term planning. 3. Empirical Performance: In our experimental section, we provide empirical evidence demonstrating that our approach surpasses previous LLM-based methodologies in the context of long-term tasks.

## 2 RELATED WORKS

**Chain of Thought for Planning.** LLMs exhibits impressive in-context learning capabilities for helping robotics planning (Ahn et al., 2022; Zeng et al., 2022; Vemprala et al., 2023). The methods used to control the LLM-agent typically involve using action and observation primitives and combining them in a chain-of-thought (CoT) manner to generate the control flow for robot planning. However, CoT reasoning has limitations as it is a static black box that relies on internal represen-

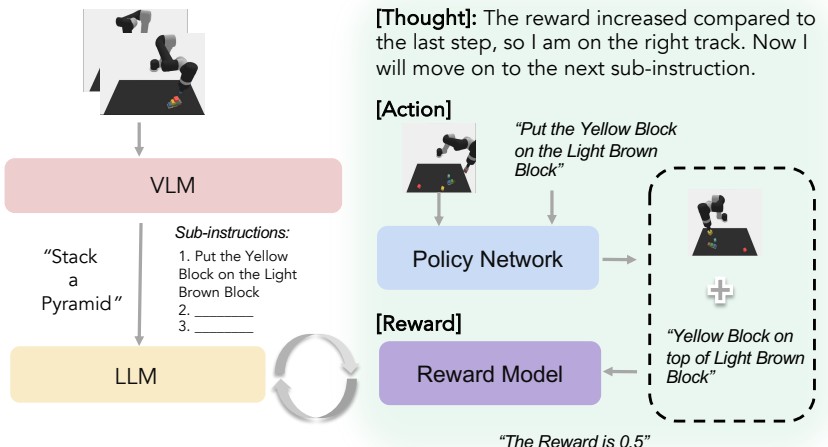

Figure 1: The overall framework of MultiReAct is designed to solve instructions that involve complex visual concepts. To achieve this, we use a Visual Language Model (VLM) to ground the visual demonstration to a sequence of language sub-instructions that begin with action primitives. Then, in each step of ReAct traces, the policy network would be called to manipulate the sub-instructions step-by-step. Additionally, we implement a reward function to check whether the current sub-instruction has been successfully executed.

tations, thereby restricting its ability to reason reactively or update its knowledge. Vanilla CoT can lead to issues such as fact hallucination and error propagation throughout the reasoning process, which hampers the performance of LLMs on long-term planning tasks. To address these limitations, ReAct (Yao et al., 2022) introduces a new approach that generates both reasoning traces and task-specific actions by utilizing tools in an interleaved manner. This enables the agent to reason from the current reasoning-acting traces, reducing hallucination and minimizing error propagation. Another approach, Toolformer Schick et al. (2023) fine-tunes LLMs with a dataset containing APIs, enabling the LLMs to autonomously determine when and how to use specific tools. This tool augmentation proves effective, especially for math reasoning and fact verifications. Additionally, multimodal module augmentation can partially alleviate the issues of hallucination and infeasible action proposals. On the other hand, multimodal module augmentation can also partially solve the problem of hallucination and prevent infeasible actions from being proposed. multimodal chain-of-Thought Zhang et al. (2023), which incorporates language and vision modalities into a two-stage framework that separates rationale generation and answer inference. Another approach, Saycan (Ahn et al., 2022) incorporates a multimodal reward function to select the most feasible plans generated by the LLM and reduce the chances of proposing actions with lower probabilities than the robots are capable of proposing.

**Failure Detection** in Embodied Planning represents a crucial task that has long been under scrutiny to enhance human trust in robotic systems. REFLECT (Liu et al., 2023) propose a system based on LLM that utilizes a hierarchical summary of multimodal robot data to provide explanations for failures and correct them. Du et al. (2023) employ language similarity as a rewarding mechanism for agents, incentivizing them to align their actions with goals suggested by a language model for effective performance. In the Cliport framework Shridhar et al. (2022), agents acquire ground truth rewards from the environment to navigate effectively towards final objectives. However, this approach is limited in real-world scenarios where obtaining ground truth rewards is often impractical. To address this challenge, our framework adopts a CLIP-based model (Radford et al., 2021) to quantify the similarity between observations in intermediate steps and sub-instructions. This integration of CLIP within the traces generated by LLMs not only facilitates the automatic generation of judgment conditions by LLMs, but also offers a versatile reward model applicable to a wide range of multimodal embodied planning tasks.

## 3 METHOD

### 3.1 PROBLEM STATEMENT

Our research presents an innovative approach to tackle the complex concepts associated with long-horizon embodied agent planning while adhering to high-level instructions. We introduce a novel framework named MultiReAct which represents a paradigm shift in the domain of chain-of-thought

(CoT) generated by LLMs. MultiReAct takes a large language model (LLM) as a planner, and inserts tools to augment its action spaces $\tilde{\mathcal{A}} = \mathcal{L} \cup \mathcal{M} \cup \mathcal{P}$. We denote $\mathcal{L}$ as the action space of LLM, $\mathcal{M}$ as the action space of VLM, $\mathcal{P}$ as the action of the policy network. Most importantly, an action $\hat{a} \in \mathcal{L}$ in the language space, which we will refer to as a thought or a reasoning trace, does not affect the external environment, thus leading to no observation feedback. Instead, a thought $\hat{a}_t$ aims to compose useful information by reasoning over the current context $c_t$, and update the context $c_{t+1} = (c_t, \hat{a}_t)$ to support future reasoning or acting. The action $a_t \in \mathcal{M} \cup \mathcal{P}$ differs from the thought $\hat{a} \in \mathcal{L}$ is the acting trace, where the acting trace in our paper refers to proposing low-level actions to interact with the embodied environment.

**Multimodal Reasoning-acting traces.** MultiReAct begins by receiving abstract and complex human instructions denoted as $l_\pi$. Alongside these instructions, we utilize a single expert demonstration $\mathcal{D}$ to illustrate the fulfillment of the high-level instruction visually. MultiAct uses a visual-language translator to transform the demonstration $\mathcal{D}$ into a dynamic sequence of actionable sub-instructions, represented as $i_0, i_1, ... i_t$. At each critical time step $t$, MultiAct orchestrates action-taking by the agent based on the contextual information provided in $c_t = \{i_1, o_1, a_1, r_1, i_2, o_2, a_2, r_2 ... i_t, o_t\}$. We denote $o_t$ as the visual observation from the environment. In our context $a_t$ is an action proposed by the language-conditioned low-level actor, $a_t = \pi(o_t, i_t)$, where $\pi$ is a pretrained actor in any form. The reward function $r_t$ serves as an essential success indicator, allowing the agent to continuously evaluate the effectiveness of sub-instruction implementation. The ultimate objective of MultiAct is to achieve high-level instruction by securing a flawless performance record in terms of reward.

## 3.2 OVERVIEW

Our framework aims to enhance the multimodal reasoning capabilities of large language models, such as GPT-3.5-Turbo-Instruct, particularly in the context of complex instruction following. We integrate several key multimodal components into the ReAct traces of these language models:

1. Visual-Language Model: This component translates actions performed in a video demonstration into step-by-step language instructions. For example, when presented with a high-level instruction like "stack a pyramid" and a corresponding video demonstration, the model generates a sequence of detailed sub-instructions (e.g., "place the red block on top of the blue block, then position the yellow block on top of the red block...").

2. Multimodal Reward Model: Leveraging CLIP, this component prompts the language model to identify and acknowledge the accomplishment of sub-goals within the task.

3. Pre-trained Actor: Our paper explores two types of actors: one based on imitation learning-based policy networks and another based on code-as-policies (Cap) actions (Liang et al., 2022). These actors play a crucial role in executing actions as part of the overall framework.

The diagram presented in Figure 1 illustrates how our framework handles long-horizon tasks with abstract instructions. The process begins with the language model invoking the video captioning model to translate the visual demonstration into a sequence of language sub-instructions. The video demonstration serves as an expert guide, consisting solely of RGB visual frames that depict how long-horizon tasks are to be executed. The sub-instructions are detailed, short-horizon actions that encompass various elements, including the necessary skills, objects involved, and more.

Each step within the ReAct of the language model involves the following components: Thought: This step is conceptually similar to previous works like Chain of Thought (CoT)(Wei et al., 2022), ReAct(Yao et al., 2022), and code-as-policies (Cap) (Liang et al., 2022). It provides free-form language prompts that offer clear interpretability and logical coherence. In essence, the language model's planner is tasked with implementing the actions specified in the sub-instructions sequentially. **Thought**: The Thought step serves as a means to observe the feedback from the previous step and determine whether the preceding actions were successfully executed, guiding the choice of subsequent actions. Notably, there is no direct interaction between the language model and the environment during this thought process. **Action**: In this step, the policy network executes low-level actions to interact with the environment, carrying out the instructions generated in the "Thought" step. **Reward**: Utilizing the CLIP model (Radford et al., 2021), this step evaluates the success status of the proposed actions by calculating a CLIP score.

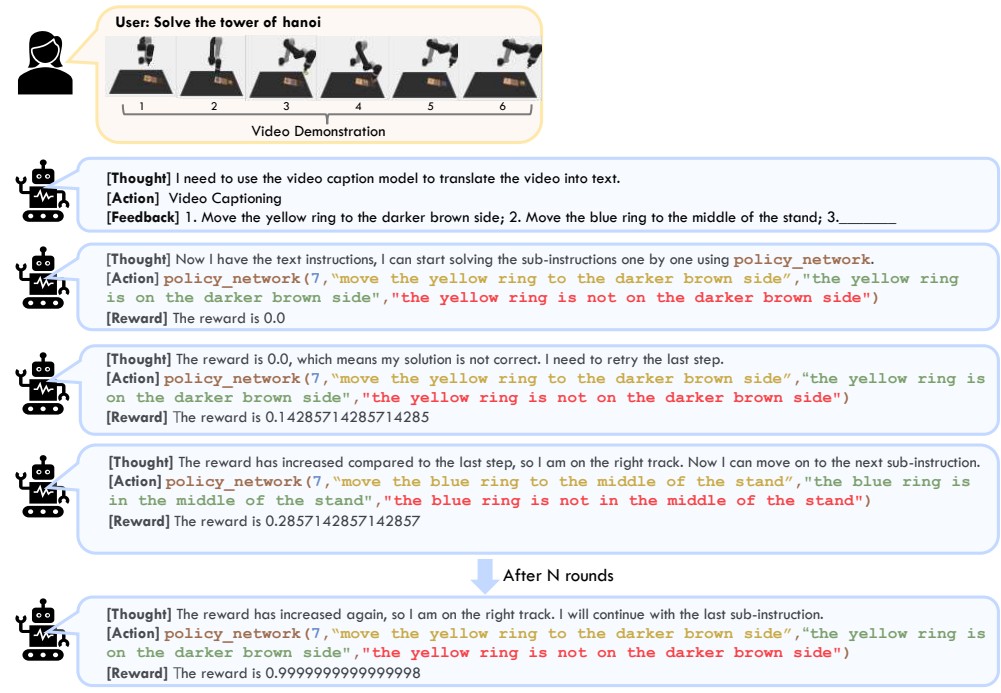

Figure 2: Examples of how MultiReAct utilizes multimodal tools to solve high-level long-horizon tasks. Within our framework, we've encapsulated the VLM, policy network, and reward function as Python APIs. The LLM planner employs an iterative execution approach for ReAct traces, involving "Thought," "Action," and "Reward," repeating this process N times until it attains a full mark reward. At each iteration, the LLM assesses whether the reward has increased. If it observes an increase, the planner utilizes the policy network to guide the next set of sub-instructions. Conversely, if there's no reward increase, the planner reattempts the previous step.

The planner iterates through the ReAct traces multiple times (typically more than $\mathcal{N}$ times, where $\mathcal{N}$ represents the number of sub-instructions provided by the Visual-Language Model) until it receives a full mark from the reward model, signifying successful task completion.

**Connect LLMs with Embodied Environment.** In our work, each tool is encapsulated as Python APIs. For example, video captioning containing a fine-tuned VLM is wrapped by the function video_captioning, the reward model containing pretrained CLIP is encapsulated by get_reward, and the policy network containing pretrained policy network is contained within the policy_network function. It's important to note that, in each ReAct step, once a tool is invoked, we utilize a Python REPL (Read-Eval-Print Loop) for interactive code execution and feedback. However, for the sake of simplicity, in our demonstrations, we have omitted the use of the Python REPL, and only demonstrated the function that has been called.

### 3.3 FINE-TUNING VISUAL LANGUAGE MODEL

**Model Architecture.** In our research, we adopt GIT as the foundational model for our visual-language framework (Wang et al., 2022). The visual encoder component of this network is based on the contrastive pre-trained model Florence (Yuan et al., 2021). This visual encoder takes raw video frames as its input and generates a compact 2D feature map. Subsequently, this feature map is flattened into a list of individual features. To further refine these features, GIT applies an additional linear layer and a layer normalization layer. This projection step transforms the image features into a D-dimensional space, which serves as the input for the text decoder. In our approach, we sample multiple frames from each rendering sequence within the Cliport dataset (Zeng et al., 2020). Each of these frames is independently encoded using the image encoder. We then introduce a learnable temporal embedding, initially initialized as zeros, and concatenate the features extracted from the

sampled frames. The text decoder, which is structured following the GPT-3 architecture Brown et al. (2020), is responsible for predicting a sequence of viable robotics actions in natural language.

**Creating Training Dataset.** In our methodology, we construct the demonstration dataset denoted as $\mathcal{D}$, which comprises pairs of expert demonstrations, long-horizon instructions, and sets of short-horizon subinstructions. Formally, $\mathcal{D} = (\zeta_1, i_{f1}, \mathcal{I}_1), (\zeta_2, i_{f2}, \mathcal{I}_2), \ldots, (\zeta_n, i_{fn}, \mathcal{I}_n)$ when we have access to n expert demonstrations $\zeta_i$, each associated with a long-horizon instruction $i_{ft}$ and a set of short-horizon sub-instructions $\mathcal{I}_t$. The visual demonstration is composed of discrete-time observation frames, represented as $\zeta_i = \{o_1, o_2, \ldots, o_n\}$. The long-horizon instruction $i_{ft}$ is a concise yet complex directive that describes the overarching task. To successfully accomplish $i_{ft}$, the agent must execute a series of sub-instructions contained within the set $\mathcal{I}_t = i_{t1}, i_{t2}, \ldots, i_{tn}$ in a step-by-step manner. During the fine-tuning process, batches of visual episodes represented as $\zeta_i$ are fed to the VLM, and the VLM is updated in a self-supervised manner based on the training data.

**Parameter Efficient Finetuning.** Having a well-trained Visual Language Model (VLM) on a general visual question-answering dataset like MS-COCO, our objective is to adapt this VLM for video question answering within embodied agent scenarios. To achieve this, we employ a parameter-efficient fine-tuning approach to strike a balance between adapting to the new task and retaining the performance on general visual-language patterns established during the initial training. In this fine-tuning process, we retain all the original parameters of the VLM while introducing additional low-rank weights to select linear layers within the attention modules. For each pairing of a visual demonstration and corresponding text, where $y_i$ represents the generated text tokens, we apply a cross-entropy loss function with label smoothing set to 0.1. The loss is computed as the average cross-entropy across all tokens within the sequence. This loss function serves the purpose of guiding the model to predict the next token in the sequence accurately, taking into account both the visual frames and the tokens generated previously.

$$loss = \frac{1}{n+1} \sum_{i=1}^{n+1} \text{CE}\left(y_i, p\left(y_i \mid \zeta, \{y_j, j = 0, \cdots, i-1\}\right)\right) \tag{1}$$

This loss function effectively encourages the model to make precise predictions for the succeeding token in the sequence, considering the visual information from the video frames and the context provided by previously generated tokens.

## 3.4 REWARD FUNCTION

At each time step, our LLM agent employs a CLIP-based reward function to evaluate the progress of the current sub-instruction. This reward function considers both positive and negative assertions, which respectively represent successful and unsuccessful scenarios. For instance, if the sub-instruction is "place the red block on top of the blue block," the positive assertion generated might be "the red block is on top of the blue block." The agent's decision to advance to the next sub-instruction is guided by an increase in reward. The choice of a positive reward hinges on which assertion better aligns with the current observation, with the reward magnitude scaled by the length of the sub-instruction sequence.

For a task comprising $N$ sub-instructions, denoted as $i_t$ at time step $t$, given the current RGB observation $o_t$, we define the positive assertion as $i_{pt}$ and the negative assertion as $i_{nt}$. The reward is calculated using the following approach:

$$r_t = \begin{cases} 0 & \text{if } \text{CLIP}(i_{pt}, o_t) < \text{CLIP}(i_{nt}, o_t) \\ \frac{1}{N} & \text{otherwise} \end{cases}$$

The whole episode will stop until the LLM agent achieves the accumulated reward 1.0.

## 3.5 LOW-LEVEL ACTOR

In our paper, we adopt two various low-level actors: 1. a pretrained Imitation-learning(IL) model Clipport (Shridhar et al., 2022), and 2. a code generation actor, Cap (Liang et al., 2022). Different from MultiReAct, low-level actors only take short-horizon language sub-instruction $i_t$ with the current observation $o_t$ of the environment, which is an image of the environment taken by the camera. The visual observation $o_t$ is a top-down orthographic RGB-D reconstruction of the scene where each pixel corresponds to a point in 3D space.

**IL-based Actor.** The semantic stream leverages a pre-trained CLIP ResNet50 (He et al., 2016) to encode RGB observations, with its decoder seamlessly integrating language features from the CLIP sentence encoder, specifically, the encoded feature of sub-instruction $i_t$.

On the other hand, the spatial stream is responsible for encoding RGB-D input, and its decoder layers are harmoniously combined with the semantic stream. The ultimate output is a high-resolution map of densely-packed pixel-wise features, which proves invaluable for making predictions related to pick and place affordances.

We denote the policy learned by Cliport as

$$\pi_{imit} = \pi(o_t, i_t) \rightarrow a_t = (\mathcal{T}_{pick}, \mathcal{T}_{place})$$

And the sub-instruction $i_t$ starts with the pre-trained skills primitives such as "move, put, pick and place, etc." The $(\mathcal{T}_{pick}, \mathcal{T}_{place})$ serves as a precise specification for the end-effector pose required for picking and placing actions.

**Code as policies.** Our paper also makes use of the same control primitives featured in Cap, as illustrated in Figure A.3. In contrast to IL-based methods, code-based policies offer several advantages: 1. Efficient Skill Adaptation: Imitation learning typically demands the collection of demonstration data for each new skill. Conversely, policy code generated by language models can efficiently repurpose existing perception and control APIs for various tasks, thus alleviating the burden of acquiring new data for each skill.

2. Precise Control: Policy code-based approaches empower the direct parameterization of low-level control APIs, enabling the specification of precise values (e.g., velocities) even in response to vague descriptions such as "faster" or "to the left." This adaptability depends on contextual cues.

Diverging from the hierarchical code generation approach discussed in Liang et al. (2022) for planning, our approach involves inserting policy code within the action section of ReAct. In this setup, the language model can iteratively invoke the policy code until the task is deemed complete, representing a distinct approach from hierarchical code generation for planning purposes.

Note that the specific action to be proposed is determined by the planner. At each step $t$, the planner assesses whether the preceding sub-instruction has been successfully executed. The confirmation of the sub-instruction's completion is contingent upon the reward received by the agent in the previous step $(t - 1)$. If the agent receives a higher reward compared to the previous step and the planning process has not yet achieved full completion (earning a perfect score of 1.0), the planner proceeds to initiate the next sub-instruction, denoted as $i_{t+1}$.

## 4 EXPERIMENTS

The objectives of our experiments are three-fold: 1. Assess the long-horizon planning capabilities of MultiReAct in comparison to other LLM-based methods. 2. We seek to verify whether augmenting the visual language model can enhance the multimodal reasoning abilities of LLMs. 3. verify whether augmenting the reward model can effectively increase the success rate in fulfilling long-horizon tasks.

### 4.1 SETUP

**Evaluation Tasks.** We evaluate the tasks proposed in CLIPort (Shridhar et al., 2022) including: short-horizon tasks: Packing Shape, Put Blocks in Bowl, and long-horizon tasks: Align Rope, Each task instance is constructed by sampling a set of objects and attributes: poses, colors, sizes, and object categories. See Appendix A.2 for detailed task information.

**Simulation Environment.** All simulated experiments are based on a Universal Robot UR5e with a suction gripper. The setup provides a systematic and reproducible environment for evaluation, especially for benchmarking the ability to ground semantic concepts like colors and object categories. The input observation is a top-down RGB-D reconstruction from 3 cameras positioned around a rectangular table: one in the front, one on the left shoulder, and one on the right shoulder, all pointing towards the center. Each camera has a resolution of $640 \times 480$ and is noiseless.

Table 1: Success rates (%) over task families in CLIPort with 50 trails per task. "S" denotes short-horizon tasks, "L" denotes long-horizon tasks.

| Task | CLIPort Based | | Cap Based | | |
| --- | --- | --- | --- | --- | --- |
| | CLIPort (gt) | MultiReAct (CLIPort) | ReAct (Cap) | Cap | MultiReAct (Cap) |
| Packing Shape (S) | 55 | **60** | 93 | **100** | 75 |
| Put Blocks in Bowl (S) | 55 | **67** | **99** | 97 | 98 |
| Align Rope (L) | 25 | **40** | - | - | - |
| Assembling kits (L) | 20 | **20** | - | - | - |
| Separating Piles (L) | **65** | 50 | - | - | - |
| Stack Block Pyramid (L) | 20 | **20** | 0 | 0 | **80** |
| Tower of Hanoi (L) | **65** | 60 | 0 | 0 | **90** |

**Evaluation Metric.** In our experiment, we adopt two metrics: 1. Success Rate: Given a set of $n$ test samples for specific robotics tasks, the success rate is determined by calculating the ratio of successful attempts to the total number of attempts. 2. Cliport Reward: We incorporate the use of ground truth reward from the Cliport environment to examine how the reward function bolsters long-horizon task planning. Before any task initiation, an oracle residing in the Cliport environment preserves the intended object pose data. As each sub-instruction reaches completion, the reward gets computed, gauging how congruous the object's current poses are to the intended ones. This scoring system provides partial credit according to the task at hand. For example, it would grant a score of 60.0 (or 3/5) for successfully packing 3 out of 5 objects as described in the instructions. Similarly, if 30 out of 56 particles are successfully maneuvered into the correct zone, it would assign a score of 53.6 (or 30/56).

## 4.2 EVALUATE LONG-HORIZON PLANNING

**Baselines.** To study the effectiveness of the MultiReAct We mainly evaluate our proposed method against the following existing approaches on the Cliport (Shridhar et al., 2022) benchmark:

- **Cliport with instruction oracle and ground truth reward.** For each task in A.2, we report the result of Cliport Shridhar et al. (2022) model that is pre-trained with 1,000 demonstrations. An oracle is employed to prompt the agent with ground truth intermediate actions in language.
- **Code as Policies** (Liang et al., 2022; Vemprala et al., 2023), a code generation method that adopt chain-of-thought for solving embodied tasks.
- **ReAct with Cap Actor** (Yao et al., 2022), we also compare with the only-language code generation Reasoning-Acting traces.

For MultiReAct we implement two versions: 1. MultiReAct with Cliport (Shridhar et al., 2022) actor 2. MultiReAct with Cap actor. For all the above-mentioned methods based on code as policies we use the same primitives in (Liang et al., 2022). For all LLM-based methods, we adopt the latest OpenAI language model GPT-3.5-turbo-0613 as the planner.

The results of the evaluation are shown in Table 1. In the context of the CLIPort-based method, our approach demonstrates competitive performance when compared to the CLIPort baseline, which directly utilizes ground truth reward signals from the environment. This highlights the efficacy of our augmented multimodal tools. Our investigation is centered on assessing how multimodal augmentation can augment the embodied planning capabilities of LLMs

**Multimoal Augmented LLM vs. text-only LLM.** When considering the Cap-based method, Cap (Liang et al., 2022) and ReAct (Yao et al., 2022) exhibit commendable performance in tasks primarily reliant on natural language understanding. However, they encounter challenges when confronted with tasks involving mathematical reasoning, spatial manipulation, and the integration of multimodal information. For instance, in the case of the Tower-of-Hanoi task, both Cap and ReAct achieved scores of 0.0, whereas MultiAct achieved an exceptional score of 99.0. This disparity can be attributed to the inherent difficulties LLMs face when handling mathematical reasoning. By incorporating a visual demonstration illustrating the solution to the Tower of Hanoi problem, we circumvent the need for the LLM to generate a precise plan and instead rely on following a sequence of sub-instructions. Furthermore, Cap and ReAct also received scores of 0.0 on the Stack-

Block-Pyramid task, revealing the current limitations of end-to-end LLMs in terms of common-sense reasoning and multimodal capabilities. The incorporation of visual demonstrations equips the LLM agent with the ability to transform intricate concepts into practical primitives through visual-language in-context learning. This augmentation plays an important role in facilitating MultiReAct's impressive achievement of a score of $80.0$ on Stack-Block-Pyramid tasks. As exemplified by the Cap model's performance in solving the Tower-of-Hanoi and Stack-Block-Pyramid tasks in Figure 4, it becomes evident that a deficiency in multimodal reasoning abilities can result in hallucinatory outcomes.

### 4.3 Ablating Reward Function.

In this section, we present an ablation study focusing on the role of the reward function in long-horizon robot planning tasks. Our objective is to investigate how the recognition and correction of failed sub-instruction executions can improve the precision of such planning tasks. To conduct this ablation analysis, we compare MultiReAct with MultiReAct without a reward function.

As illustrated in Figure 3, we have observed that the reward function significantly enhances performance, particularly in tasks such as Align-Rope and Separating-Piles. In the case of the Align-Rope task, MultiReAct (without the reward function) achieves a CLIP reward score of 13, while MultiReAct with the reward function achieves a considerably higher score of 55.

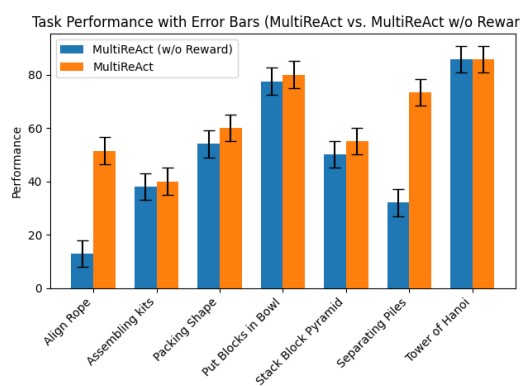

Figure 3: Performance comparison of MultiReAct and MultiReAct w/o reward function on CLIPort.

Furthermore, in the Separating-Piles task, MultiReAct (without the reward function) attains a CLIP reward score of 32, whereas MultiReAct with the reward function reaches an even more impressive score of 73.

The phenomenon of MultiReAct to be weak on these tasks can be partially attributed to the absence of a reward model. Without such a model, the LLM agent tends to consider each executed step as completely correct. In contrast, when a reward model is incorporated, the LLM agent gains the ability to identify instances of failed executions and subsequently retry those actions.

## 5 Conclusion

In conclusion, we've introduced MultiReAct, a simple yet powerful approach framework to enhance the multimodal reasoning and long-horizon planning capabilities of large language models. Our extensive experiments spanned various long-horizon embodied planning tasks involving uncommon objects, including rope alignment, block pyramid stacking, Tower of Hanoi solving, and block pile separation, among others. Despite its simplicity, our method consistently outperforms previous language model-based approaches when applied to complex long-horizon tasks demanding advanced multimodal reasoning skills. Our comprehensive ablation studies underscore the significance of augmenting both the visual language model and the reward model in elevating long-horizon embodied planning performance. We've demonstrated that our framework exhibits strong in-task generalization, effectively adapting to shifting object layouts and attributes within specific scenarios. However, we acknowledge a limitation regarding cross-task generalization, particularly in the context of unseen tasks, which heavily relies on the capabilities of the visual language model. Moreover, it would be worthwhile to investigate the potential of scaling up the MultiReAct on more complex planning tasks through multi-task training. Combining it with complementary paradigms such as reinforcement learning could lead to the development of more resilient agents, thereby unlocking the full potential of LLMs for diverse applications in embodied AI.

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

## A   APPENDIX

### A.1   REPRODUCIBILITY STATEMENT

Our primary experiments employ the following models: 1. For the Large Language Model planner, we utilize GPT-3.5-turbo-0613. 2. In the case of the Video Language Model, we employ GIT (Wang et al., 2022). However, since our main focus lies in long-horizon embodied planning rather than visual-to-language generation, readers can also explore the latest and more robust visual language model which may exhibit superior performance.

### A.2   CLIPORT DATASET

As for the benchmark for evaluating our proposed method with the baselines, we use the following datasets from Cliport (Shridhar et al., 2022).

**1. Assembling Kits   Task:** Precisely place each specified shape in the specified hole following the order prescribed in the language instruction generated by the LLM-based agent at each timestep. This is one of the hardest tasks in the benchmark requiring precise placements of shapes of randomized colors and grounding spatial relationships. Each task instance contains 5 shapes and a kit with randomized poses. **Goal:** assembling all the kits to the corresponding holes. **Success Metric:** The pose of each shape matches the specified hole at the correct timestep. The final score is the total number of shapes that were placed in the correct pose at the correct timestep, divided by the total number of shapes in the scene (always 5).

**2. Align Rope   Task:** Manipulate a deformable rope to connect its end-points between two corners of a 3-sided square. There are four possible combinations for aligning the rope: "front left tip to front right tip". "front right tip to back right corner", "front left tip to back left corner", and "back right corner to back left corner". Here 'front' and 'back' refer to canonical positions on the 3-sided square. The poses of both the rope and 3-sided square are randomized for each task instance. **Objects:** All align-rope instances contain a rope with 20 articulated beads and a 3-sided square. **Success Metric**: The poses of all beads match the line segments between the two correct sides.

**3. Stack Block Pyramid.   Task:** Build a pyramid of colored blocks in a color sequence specified through the step-by-step language instructions generated by the LLM-based agent. Each task contains 6 blocks with randomized colors and 1 rectangular base, all initially placed at random poses. **Goal:** stack a pyramid by 6 blocks and 1 rectangular base. **Success Metric:** The pose of each block at the corresponding timestep matches the specified location. The final score is the total number of blocks in the correct pose at the correct timestep, divided by the total number of blocks (always 6).

**4. Towers of Hanoi.   Task:** Move the ring to the specified peg in the LLM-based agent-generated language instruction at each timestep. The sequence of ring placements is always the same, i.e. the perfect solution to three-ring Towers of Hanoi. This task can be solved without using colors by just observing the ring sizes. However, it tests the agent's ability to ignore irrelevant concepts to the task (color in this case). The task involves precise pick and place actions for moving the rings from peg to peg. **Goal:** Solve the tower of Hanoi via moving 3 rings (small, medium, and big) across 1 peg base. **Success Metric:** The pose of each ring at the corresponding timestep matches the specified peg location. The final score is the total number of correct ring placements, divided by the total steps in the perfect solution (7 for three-ring Towers of Hanoi).

**5. Put Blocks in Bowl.   Task:** Place all blocks of a specified color in a bowl of the specified color. Each bowl fits just one block and all scenes contain enough bowls to achieve the goal. Each task instance contains several distractor blocks and bowls with randomized colors. The solutions to this task are multi-modal in that there could be several ways to place the blocks specified in the language goal. This task does not require precise placements and mostly tests an agent's ability to ground color attributes. **Goal:** to place blocks with certain colors on the bowls with certain colors **Success Metric:** All blocks of the specified color are within the bounds of a bowl of the specified color. The final score is the total number of correct blocks in the correct bowls, divided by the total number of relevant color blocks in the scene.

**6. Packing Shapes.**   **Task:** Place a specified shape in the brown box. Each task instance contains 1 shape to be picked along with 4 distractor shapes. The shape colors are randomized but have no relevance to the task. This task does not require precise placements and is mostly a test of the agent's semantic understanding of arbitrary shapes. **Goal:** To pack the required shapes to the brown box **Success Metric**: The correct shape is inside the bounds of the brown box.

**Task:** Sweep the pile of blocks into the specified zone. Each scene contains two square zones: one relevant to the task, another as a distractor. The pile and zones are placed at random poses on the table.

**5. Separating Piles.**   **Task:** Sweep the pile of blocks into the specified zone. Each scene contains two square zones: one relevant to the task, another as a distractor. The pile and zones are placed at random poses on the table. **Objects:** A pile of colored blocks and two squares. **Success Metric:** All blocks are inside the bounds of the specified zone. The final score is the total number of blocks inside the correct zone, divided by the total number of blocks in the scene.

### A.3   LOW-LEVEL POLICY NETWORK

**Code as Policies.** The primitives listed below are used in Cap (Liang et al., 2022). Note that we also used the same control and observation primitives without introducing new functions.

```python
class LMP_wrapper():

  def __init__(self, env, cfg, render=False):
    self.env = env
    self._cfg = cfg
    self.object_names = list(self._cfg['env']['init_objs'])

    self._min_xy = np.array(self._cfg['env']['coords']['bottom_left'])
    self._max_xy = np.array(self._cfg['env']['coords']['top_right'])
    self._range_xy = self._max_xy - self._min_xy

    self._table_z = self._cfg['env']['coords']['table_z']
    self.render = render

  def is_obj_visible(self, obj_name):
    return obj_name in self.object_names

  def get_obj_names(self):
    return self.object_names[::]

  def denormalize_xy(self, pos_normalized):
    return pos_normalized * self._range_xy + self._min_xy

  def get_corner_positions(self):
    unit_square = box(0, 0, 1, 1)
    normalized_corners = np.array(list(unit_square.exterior.coords))[:4]
    corners = np.array(([self.denormalize_xy(corner) for corner in
        normalized_corners]))
    return corners

  def get_side_positions(self):
    side_xs = np.array([0, 0.5, 0.5, 1])
    side_ys = np.array([0.5, 0, 1, 0.5])
    normalized_side_positions = np.c_[side_xs, side_ys]
    side_positions = np.array(([self.denormalize_xy(corner) for corner
        in normalized_side_positions]))
    return side_positions

  def get_obj_pos(self, obj_name):
    # return the xy position of the object in robot base frame
    return self.env.get_obj_pos(obj_name)[:2]
```

```python
41    def get_obj_position_np(self, obj_name):
42      return self.get_pos(obj_name)
43
44    def get_bbox(self, obj_name):
45      # return the axis-aligned object bounding box in robot base frame
              (not in pixels)
46      # the format is (min_x, min_y, max_x, max_y)
47      bbox = self.env.get_bounding_box(obj_name)
48      return bbox
49
50    def get_color(self, obj_name):
51      for color, rgb in COLORS.items():
52        if color in obj_name:
53          return rgb
54
55    def pick_place(self, pick_pos, place_pos):
56      pick_pos_xyz = np.r_[pick_pos, [self._table_z]]
57      place_pos_xyz = np.r_[place_pos, [self._table_z]]
58      pass
59
60    def put_first_on_second(self, arg1, arg2):
61      # put the object with obj_name on top of target
62      # target can either be another object name, or it can be an x-y
              position in robot base frame
63      pick_pos = self.get_obj_pos(arg1) if isinstance(arg1, str) else arg1
64      place_pos = self.get_obj_pos(arg2) if isinstance(arg2, str) else arg2
65      self.env.step(action={'pick': pick_pos, 'place': place_pos})
66
67    def get_robot_pos(self):
68      # return robot end-effector xy position in robot base frame
69      return self.env.get_ee_pos()
70
71    def goto_pos(self, position_xy):
72      # move the robot end-effector to the desired xy position while
              maintaining same z
73      ee_xyz = self.env.get_ee_pos()
74      position_xyz = np.concatenate([position_xy, ee_xyz[-1]])
75      while np.linalg.norm(position_xyz - ee_xyz) > 0.01:
76        self.env.movep(position_xyz)
77        self.env.step_sim_and_render()
78        ee_xyz = self.env.get_ee_pos()
79
80    def follow_traj(self, traj):
81      for pos in traj:
82        self.goto_pos(pos)
83
84    def get_corner_positions(self):
85      normalized_corners = np.array([
86          [0, 1],
87          [1, 1],
88          [0, 0],
89          [1, 0]
90      ])
91      return np.array(([self.denormalize_xy(corner) for corner in
              normalized_corners]))
92
93    def get_side_positions(self):
94      normalized_sides = np.array([
95          [0.5, 1],
96          [1, 0.5],
97          [0.5, 0],
98          [0, 0.5]
99      ])
100     return np.array(([self.denormalize_xy(side) for side in
              normalized_sides]))
```

```python
101
102  def get_corner_name(self, pos):
103    corner_positions = self.get_corner_positions()
104    corner_idx = np.argmin(np.linalg.norm(corner_positions - pos,
           axis=1))
105    return ['top left corner', 'top right corner', 'bottom left corner',
           'botom right corner'][corner_idx]
106
107  def get_side_name(self, pos):
108    side_positions = self.get_side_positions()
109    side_idx = np.argmin(np.linalg.norm(side_positions - pos, axis=1))
110    return ['top side', 'right side', 'bottom side', 'left
           side'][side_idx]
```

Listing 1: The Cap primitives

## A.4  LLM PROMPTS

The following text illustrates how we primarily prompt the large language model.

---

Assume you are a Robotics Agent that can use Tools to solve a task. you are allowed to use the following tools: {tool_description} You are allowed to call the following actions:

1. `video_captioning`, given a high-level instruction you should parse it to low-level instructions by using Python REPL to call this function, use `video_captioning()` as Action Input of Python REPL, do not add additional parameters.

2. `policy_network`, given a low-level instruction you should use the policy network to solve the instruction by using Python REPL to call this function, use `policy_network(num_sub_inst, lang_goal, positive_text, negative_text)` as Action Input of Python REPL. For example, if instructions contain 6 sub-instructions, and the sub-instruction is "put the blue block on the lightest brown block," then you should call `policy_network(6, ``put the blue block on the lightest brown block", ``the blue block is on the lightest brown block", ``the blue block is not on the lightest brown block")` as Action Input of Python REPL.

For each time step, you should use the following format to describe your solution. Do not generate all the steps at once; you should generate the steps one by one:

**Thought:** Comment on what you want to do next

**Action:** The action to take, exactly one element of {tool_names}

**Action Input:** The input to the action, either `video_captioning()` or `policy_network(``sub_instruction")`. Do not include Action or Observation in the Action Input.

(Repeat the above Thought/Action/Action Input N times, use it until you are sure you finished the task)

In the first step, you have to use the video caption model to translate the video into text, by using Python REPL to call `video_captioning()` as Action Input of Python REPL. Then you have to use the policy network to solve the sub-instructions one by one, by using Python REPL to call `policy_network(``sub_instruction")` as Action Input of Python REPL.

**Begin! Instruction:** {instruction}

**Thought:** {previous_responses}

---

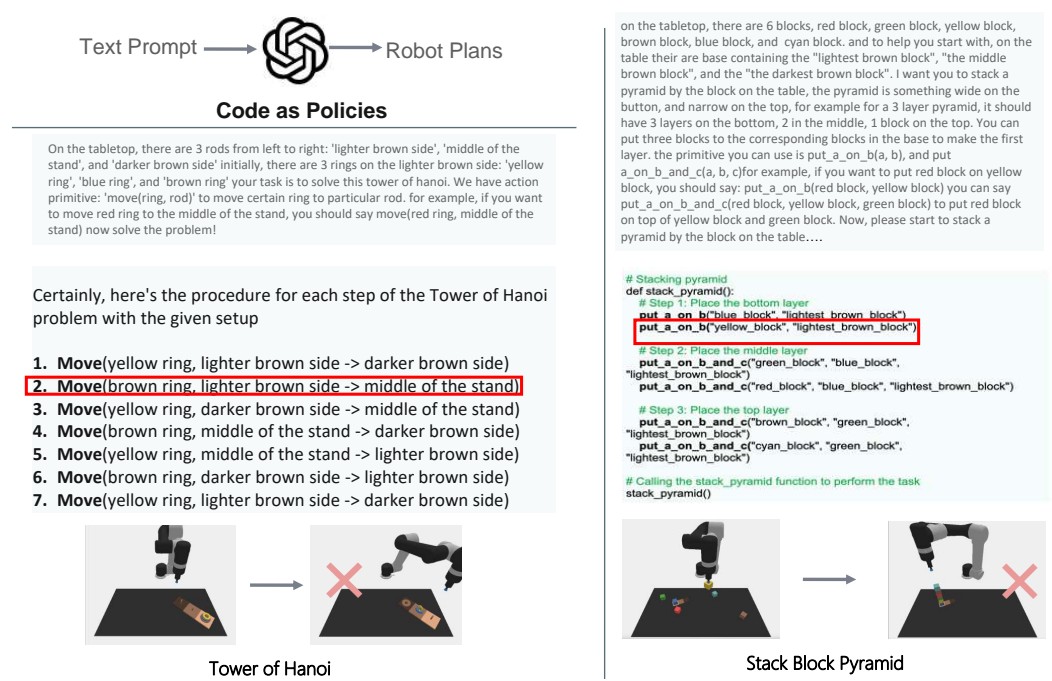

Figure 4: Cap (Liang et al., 2022) hallucination on the stack-block-pyramid task and tower of Hanoi task, the red boxes indicate the places where the agent starts to make faults.

## A.5 ADDITIONAL RESULTS

**LLM hallucinates on long-horizon planning.**

Figure 4 illustrates how Cap fails on the long-horizon planning tasks: Tower-of-Hanoi and Stack-Block-Pyramid. The red boxes indicate the step LLM starts to hallucinate.

**Example of MultiReAct solve Stack-Block-Pyramid task.** The following text illustrates how MultiReAct generated the ReAct trace to solve the stack-block-pyramid puzzle.

```
Thought:
I will start by using the video caption model to translate the video
    into text.
Action: Python REPL
Action Input: video_captioning()
Feedback: 1. put the cyan block on the lightest brown block 2. put the
    yellow block on the middle brown block 3. put the blue block on the
    darkest brown block 4. put the green block on the cyan and yellow
    blocks 5. put the brown block on the yellow and blue blocks 6. put
    the red block on the green and brown blocks.

Thought:
Now that I have the low-level instructions, I will use the policy
    network to solve each sub-instruction one by one.
Action: Python REPL
Action Input: policy_network(6, "put the cyan block on the lightest
    brown block", "the cyan block is on the lightest brown block", "the
    cyan block is not on the lightest brown block")
Reward: the reward is  0.16666666666666666

Thought:
The reward has increased compared to the last step, so I am on the right
    track. Now I will move on to the next sub-instruction.
Action: Python REPL
```

```
16 Action Input: policy_network(6, "put the yellow block on the middle
       brown block", "the yellow block is on the middle brown block", "the
       yellow block is not on the middle brown block")
17 Reward: the reward is  0.3333333333333333
18
19 Thought:
20 The reward has increased compared to the last step, so I am on the right
       track. Now I will move on to the next sub-instruction.
21 Action: Python REPL
22 Action Input: policy_network(6, "put the yellow block on the middle
       brown block", "the yellow block is on the middle brown block", "the
       yellow block is not on the middle brown block")
23 Reward: the reward is  0.3333333333333333
24
25 Thought:
26 The reward remains unchanged, which means my solution is not correct. I
       need to retry the last step.
27 Action: Python REPL
28 Action Input: policy_network(6, "put the blue block on the darkest brown
       block", "the blue block is on the darkest brown block", "the blue
       block is not on the darkest brown block")
29 Reward: the reward is  0.5
30
31 Thought:
32 The reward has increased again, so I will continue to the next
       sub-instruction.
33 Action: Python REPL
34 Action Input: policy_network(6, "put the green block on the cyan and
       yellow blocks", "the green block is on the cyan and yellow blocks",
       "the green block is not on the cyan and yellow blocks")
35 Reward: the reward is  0.6666666666666666
36
37 Thought:
38 The reward has increased again, so I will continue to the next
       sub-instruction.
39 Action: Python REPL
40 Action Input: policy_network(6, "put the brown block on the yellow and
       blue blocks", "the brown block is on the yellow and blue blocks",
       "the brown block is not on the yellow and blue blocks")
41 Reward: the reward is  0.8333333333333333
42
43 Thought:
44 The reward has increased again, so I will continue to the final
       sub-instruction.
45 Action: Python REPL
46 Action Input: policy_network(6, "put the red block on the green and
       brown blocks", "the red block is on the green and brown blocks",
       "the red block is not on the green and brown blocks")
47 Reward: the reward is  0.9999999999999999
```

Listing 2: The procedure of how MultiReAct solve stack block pyramid problem

