# OpenReview forum: "MultiReAct: Multimodal Tools Augmented Reasoning-Acting Traces for Embodied Agent Planning"
_ICLR.cc/2024/Conference — Submitted to ICLR 2024_

### Official Review · Reviewer_LrB8 · 2023-10-29

**Soundness:** 3 good
**Presentation:** 3 good
**Contribution:** 3 good
**Rating:** 8
**Confidence:** 3

**Summary:**

Given an instruction, LLMs can generate a sequence of sub-instructions to follow the instruction, but the generated sub-instructions are often too abstract and lack intricate visual descriptions. To address this problem, the authors propose a novel paradigm, taking a video demonstration as well as a natural language instruction as input, to generate sub-instructions with more detailed information.

To get feedback from the real environment, the authors also propose to use CLIP-based reward to judge whether the sub-instruction is executed successfully.

**Strengths:**

While LLM has the ability to generate abstract plans (in the form of a sequence of sub-instructions in natural language) given natural language instruction, the problem is that LLMs are not grounded in real environments. How to make LLMs-based planning more grounded? I think the authors give a very effective solution, introducing video demonstrations to provide more details about the real environments, enabling VLMs to generate more grounded sub-instructions, and introducing feedback from environments to judge whether the sub-instructions are executed successfully.

**Weaknesses:**

see question

**Questions:**

I am just curious if the CLIP reward can be leveraged to finetune the low-level policy.

---

> ### Author Response · Authors · 2023-11-20
>
> We sincerely thank the reviewer for their constructive feedback and insightful question regarding the potential use of CLIP reward in fine-tuning the low-level policy. We appreciate your recognition of our approach to ground LLM-based planning in real environments using video demonstrations and CLIP-based reward. Your understanding of the significance of our method in enhancing the detail and contextual relevance of sub-instructions is encouraging.
>
> ```
> Q1: I am just curious if the CLIP reward can be leveraged to finetune the low-level policy.
> ```
> Yes, the potential fine-tuning process could be:
> **Generating Feedback**: After the low-level policy executes an action, the result (captured as an image or video frame) is fed into CLIP along with the textual description of the intended action or task. CLIP then evaluates how well the result aligns with the description.
> **Reward Mechanism**: Based on CLIP's evaluation, a reward signal can be generated. If CLIP finds a high correlation between the action's outcome and the textual task description, a positive reward is given. Conversely, a low correlation results in a lower reward or a penalty.
> **Policy Adjustment**: This reward signal is used to fine-tune the low-level policy's neural network.  RL algorithms such as PPO can be employed here, where the policy network's parameters are adjusted to maximize the received rewards over time.
> Iterative Improvement: Over multiple iterations, the low-level policy learns to generate actions that increasingly align with the task descriptions, as validated by CLIP. The policy becomes more adept at interpreting instructions in the context of the visual environment it operates in.
>
> We believe this integration would be significantly contribute to the field of embodied AI. Once again, thank you for your valuable feedback and the opportunity to consider this exciting direction for our research.

---

> > ### Comment · Reviewer_LrB8 · 2023-11-21
> > **One More Question**
> >
> > I just found a related paper “Prompting Decision Transformer for Few-Shot Policy Generalization”. Therefore, I actually have one more question. Their work uses demonstrations as prompts, which is quite similar to the way you use video demonstrations as input. Could the author explain the novelty against the forementioned paper?

---

> > > ### Author Response · Authors · 2023-11-21
> > > **Response to Query on Novelty Compared to 'Prompting Decision Transformer'**
> > >
> > > That's a great question, we appreciate your bringing up the "Prompting Decision Transformer for Few-Shot Policy Generalization" paper. It provides us with an excellent opportunity to clarify the unique aspects and novel contributions of our work. Our approach differs from and advances beyond the methodology presented in the mentioned paper in the following key ways:
> > >
> > > **Transformation of Video Demonstrations into Semantic-Level Planning**:
> > >
> > > Unlike the Prompting Decision Transformer (Prompt-DT) approach, which inputs demonstrations directly as prompts requiring the model to learn related self-attention, our model transforms video demonstrations into semantic-level planning. This transformation results in human-understandable plans that can be seamlessly integrated with code generation, offering superior expandability. Our approach ensures that the planning is not just a replica of the demonstration but a semantically rich interpretation that aligns with human logic and understanding.
> > >
> > > **Independence from Action Labels in Demonstrations**:
> > >
> > > A significant distinction is that the Prompt-DT relies on demonstrations that provide action labels, whereas our multimodal model does not require action labels in the demonstrations. This feature is particularly beneficial as, in many real-world scenarios, obtaining action labels for demonstrations can be challenging and costly. Video demonstrations are often more readily available and easier to acquire. Our method leverages this availability, making it more practical and applicable in diverse settings where action labels might not be easily accessible.

---

### Official Review · Reviewer_Ufhp · 2023-11-01

**Soundness:** 3 good
**Presentation:** 2 fair
**Contribution:** 2 fair
**Rating:** 5
**Confidence:** 4

**Summary:**

This paper presents a method to incorporate 1) video captioning to generate textual instructions from visual demonstrations, 2)  a reward model based on multimodal similarity of the visual input observations and generated instructions, and 3) incorporating the ReAct framework for EmbodiedAI via the incorporation of a low-level policy.

**Strengths:**

- the proposed MultiReAct framework connects several existing frameworks such as CoT, ReAct, Cap, and VLMs under a unified general framework for improving planning via LLMs in EmbodiedAI scenarios.
- The idea of using an environment-independent general-purpose reward model is very appealing and makes the presented framework general.
- The improvements over long-horizon tasks are significant compared to Cap and ReAct agents

**Weaknesses:**

**missing details**
paper lacks several details in the presentation of model's module, training/finetuning, etc. Please see questions.


**presentation errors**
There are some disagreeing numbers and confusing statements in the paper. Please see questions.

**validity of empirical results**
there seems to be no repetition with various random seeds to measure variation across different runs, also no significance test as well as no use of more recent measures such as IQM[1] were used to validate the reported results are statistically meaningful.

**limitation of empirical evidence**
Only one environment was used to demonstrate the effectiveness of the presented framework which is framed as a very generalistic approach. This significantly limits support for the presented framework.

[1] Agarwal, Rishabh, et al. "Deep reinforcement learning at the edge of the statistical precipice." Advances in neural information processing systems 34 (2021): 29304-29320.

**Questions:**

**missing details**
- in the comparisons to Cliport, what exactly, module-by-module, and in terms of input and outputs, is the difference between the Cliport model and MultiReAct that uses a Cliport actor? Doesn't the MultiReAct that use a fully trained Cliport policy? if so, the trained policy has already been trained with the gt reward, hence, has been given more information than Cliport (data it had access to during pretraining, as well as additional pretraiend models). If this is the case, then the expectation is that MultiReAct should significantly surpass Cliport. I think a detailed breakdown between the two models are needed to clarify the benefits of MultiReAct.

-   Providing information on the model's architecture with notations on which parts are finetuned and with what data, and which parts are frozen, will add more clarity. At the moment this information is scattered across the paper and hard for the reader to get the bigger picture. To my understanding, the only finetuned part is the VLM, finetuned on the dataset introduced in Section 3.3.

**presentation errors**
- In Table 1, MultiReAct (Cap) for Tower of Hanoi has SR of 90 but in text in Section 4.2 it has been stated 99.0.
- The y-axis in Figure 3 is labeled as Performance, without mentioning what that is. SInce two measures of evaluation have been introduced (SR and Cliport reward), I assume that refers to Cliport reward? If so, why in the text in Section 4.3 while referring to Figure 3, numbers for Clip Reward are stated and not Cliport reward?


**validity of empirical results**
there seems to be no repetition with various random seeds to measure variation across different runs, also no significance test as well as no use of more recent measures such as IQM[1] were used to validate the reported results are statistically meaningful. Please provide either significant tests or use IGM.

**limitation of empirical evidence**
The framework has been presented as a generalist approach to embodied AI. While there are many available benchmarks for Embodied AI such as Habitat, MineDojo, etc, this paper can be strengthened by extending empirical evidence to such environments, or rephrase statements to match the level of generality that the empirical evaluations can support.

[1] Agarwal, Rishabh, et al. "Deep reinforcement learning at the edge of the statistical precipice." Advances in neural information processing systems 34 (2021): 29304-29320.

---

### Official Review · Reviewer_L9T3 · 2023-11-03

**Soundness:** 2 fair
**Presentation:** 3 good
**Contribution:** 2 fair
**Rating:** 3
**Confidence:** 4

**Summary:**

In this work, the authors propose a framework for embodied agent planning based on a system of vision-language model, large-language model, and pre-trained low-level actors. Specifically, their approach, named MultiReAct, utilizes a parameter-efficient adaptation of a pre-trained VLM  for video captioning (converting input instruction and a visual demo into sub-instructions) and empowers the ReAct framework which interleaves reasoning (via text) and acting (via low-level policies) with multimodal input signals and reward signals (via CLIP). The authors show its superior performance on both short and long-horizon planning tasks than previous LLM-based methods.

**Strengths:**

+ The framework is very intuitive and well-motivated
+ The method is straightforward, simple and clean

**Weaknesses:**

- The novelty is limited as it essentially adds multimodal rewards with CLIP, which is more or less an existing idea (see, e.g., [1]), and video captioning for converting instruction and video demos into sub-instructions to the existing ReAct framework.
- Due to the point above, I would expect stronger experiment results in terms of the broadness of the tasks. The authors evaluate their framework only on the CLIPort tasks, which are relatively simple (e.g., the suction-based gripper abstracts away most of the low-level actions). Tasks involving more low-level controls (e.g., object manipulations) should be evaluated or at least discussed, where whether the CLIP or any existing 2D-based VLM is good enough for providing useful reward signals remains a question. For instance, the CALVIN benchmark. Alternatively, other popular embodied agent planning tasks involving only high-level actions such as ALFRED should be evaluated or at least discussed to give a better sense of the applicability of this framework.

[1] Zero-Shot Reward Specification via Grounded Natural Language

**Questions:**

See "weakness". I am willing to increase my score after seeing the author's response.

---

> ### Author Response · Authors · 2023-11-21
> **Reply to  Reviewer L9T3**
>
> We appreciate the reviewer's observations and critique regarding the perceived novelty of our approach and the breadth of our experimental evaluation. We understand the importance of distinguishing our work from existing literature and demonstrating its practical applicability in a range of scenarios.
>
> > w1: novelty
>
> **Multimodal Rewards with CLIP Integration**:
>
> While integrating multimodal rewards with CLIP might be an existing idea. In our methodology, the Large Language Model (LLM) planner interacts with the CLIP model to dynamically generate both negative and positive conditions. This process is crucial for determining whether a sub-instruction has been successfully executed. Unlike static reward systems, our approach adapts the reward criteria based on the specific context of each task, leading to more nuanced and accurate evaluations.
>
> >w2: suction-based gripper abstracts away most of the low-level actions
>
> Clarifying Research Motivation and Focus:
>
> 1. Emphasis on Embodied Planning:
> Our primary objective is to enhance embodied planning capabilities within AI systems. This focus is specifically geared towards understanding and improving how AI systems can develop and execute plans based on abstract instructions. Our research is less about physical manipulation and more about the cognitive process of planning in response to given tasks.
>
> 2. Scope of Current Experiments:
> The reason we chose CLIPort tasks for our initial evaluation is that they provide a controlled environment to test our framework's planning capabilities. While these tasks may appear simple due to the abstraction of low-level actions, they are quite effective in demonstrating the efficacy of our approach in generating and executing plans based on multimodal inputs.
>
> 3. Future Directions and Broader Applicability:
> We acknowledge the importance of evaluating our framework in more diverse scenarios, including tasks that involve more intricate object manipulations (as in the CALVIN benchmark) or high-level action planning (such as in ALFRED). While these are valuable areas for future research, they were beyond the initial scope of this study. Our immediate goal was to establish a foundational understanding of how multimodal tools can enhance the planning process in AI systems.

---

### Official Review · Reviewer_ftHn · 2023-11-05

**Soundness:** 2 fair
**Presentation:** 2 fair
**Contribution:** 2 fair
**Rating:** 3
**Confidence:** 3

**Summary:**

This work proposes the MultiReAct method, which consists of 3 main components: 1. a vision language model for captioning demonstrations in image forms into text forms; 2. A large language model for orchestrating the video captioning model and 3. A CLIP-based discriminator for evaluation of sub-task progress.

Concretely, the VLM is fine-tuned with LoRA on a dataset of demonstrations containing the long-horizon instruction, sub-instructions, and demonstrated image observations.

The CLIP-based reward function (discriminator) takes in an image observation, a positive proposition, and a negative proposition. It compares the similarity score to determine whether the current observation has completed a particular sub-instruction. Note that here the positive and negative propositions are generated by the LLM.

Finally, the LLM is provided access to trainer VLM and actor modules. It is instructed to parse the visual demonstration into textual sub-instructions and then execute them sequentially. The CLIP-based discriminator provides the signal for the LLM to determine whether the actor has finished the current sub-instruction and can proceed to the next one.

The authors experimented with CLIPort and Code-as-Policy actors on a suite of short and long-horizon tasks. The results of CLIPort-based methods show that the MultiReAct performs similarly to CLIPort although it doesn't have access to the ground-truth reward signals. Further ablation studies also prove the utility of the CLIP-based reward. Results on Cap-based methods show that MultiReAct combined with Cap enables successful learning of long-horizon tasks.

**Strengths:**

- The grounding problem with the existing LLM-based planner that the authors address is well-motivated.
- The authors did a good job explaining the overall pipeline and its components.
- The method makes sense. As shown the CLIP-based reward seems effective.

**Weaknesses:**

- The experiment setting is missing some key details (see questions).
- I'm not sure if the main experiment comparison is fair. For example, do ReAct (Cap) and Cap also have access to demonstrations? These may be necessary for long-horizon tasks. Also I believe CLIPort doesn't get demonstration at test time. The authors should elaborate on the experiment settings.
- The LLM is instructed to call the VLM and then iteratively call the actor and reward model. This is probably achievable with a simple program. The use of LLM here might be an overkill.

**Questions:**

- Maybe a typo for the word "excitability"?
- How many demonstrations are needed to fine-tune the VLM? How many are used to train CLIPort?
- Is the VLM fine-tuned for each individual task or on all tasks? Does it generalize to unseen tasks?
- How are the Stack Block Pyramid and Tower of Hanoi tasks randomized? As far as I can tell, MultiReAct will try to replicate exactly what has been done in the demonstration. If this is the case, would the task fail if the colors and initial ring locations are swapped?
- Why do you use CLIP reward score as performance measure in section 4.3 but not the ground-truth reward?

---

> ### Author Response · Authors · 2023-11-20
> **Reply to Reviewer ftHn (1/2)**
>
> Thanks for your comments. We address your major concerns one by one.
> > W1: Experiment Fairness
>
> To ensure the fairness of comparison we make the following setting
> - **CLIPort:** During test time, CLIPort does not directly access video demonstrations. However, it is equipped with an oracle feature that accurately provides it with the necessary sub-instructions at each time step. This setup is intended to simulate a scenario where CLIPort can rely on high-quality, contextual guidance to execute tasks, without the direct need for visual demonstrations.
> - **CaP and ReAct:** Both CaP (Comprehension and Planning) and ReAct (Cap) systems are provided with general instructions on how to fulfill the tasks. These instructions are comparable to what a human operator might receive.  For example, we tell LLMs ``a pyramid is something wide at the bottom and narrow on the top, for a 3 layer pyramid, it should have 3 blocks on the bottom, 2 blocks in the middle and 1 on the top.``  For a detailed view of these instructions, we refer you to Figure 4 in our paper.
> - **MultiReAct**: Unlike CLIPort, which utilizes an oracle feature for sub-instruction guidance, or CaP and ReAct, which access general task instructions, MultiReAct relies exclusively on multimodal reasoning. This approach integrates visual data with abstract robot instructions, enabling the LLMs to infer appropriate actions without the need for explicit oracles or detailed guidelines.
>
> > W2: Is LLMs overkill?
>
> The LLMs is suitable for the planning tasks. For the following reasons:
> 1. **Complex Decision-Making and Integration:**
>     The LLM's role extends far beyond that of a mere sequential call dispatcher. It is integral in orchestrating complex decisions that involve the integration of multimodal inputs and the interpretation of nuanced instructions within varying contexts. This sophisticated level of decision-making and integration, essential for our method, is beyond the capabilities of a simple program.
> 2. **Advanced Natural Language Processing:**
>     LLMs excel in understanding and generating human-like text, a key feature that enhances the intuitiveness and effectiveness of interactions, particularly with abstract or complex instructions. Moreover, the plans generated by LLMs are marked by superior interpretability, facilitating clearer communication and execution of tasks.
> 3. **Scalability and Versatility:**
>     The LLM-based planning pipeline offers remarkable scalability, adaptable to a wide array of robotics tasks. Unlike fixed programs, LLMs can adjust to new tasks with minimal modifications, demonstrating their versatility and applicability across diverse scenarios.
> 4. **Enhancing Reward Model Judgments:**
>      LLMs contribute significantly to setting the judgment criteria for reward models. They generate context-specific positive and negative conditions, guiding the reward models to accurately assess whether a sub-instruction has been successfully fulfilled.

---

> ### Author Response · Authors · 2023-11-20
> **Reply to Reviewer ftHn (2/2)**
>
> > Q1: How many demonstrations are needed to fine-tune the VLM? How many are used to train CLIPort?
>
> Both our VLM in the MultiReAct system and CLIPort are trained on a dataset comprising 100$T$ demonstrations. $T$ denotes the number of tasks.
>
>
> > Q2: Is the VLM fine-tuned for each individual task or on all tasks? Does it generalize to unseen tasks?
>
> Our VLM is fine-tuned across a broad spectrum of tasks, rather than on individual tasks.
> **Incorporating Randomization:** We introduce randomization in the layout and arrangement within each task type. This strategy is crucial for the VLM to handle variations and complexities within the same task domain, ensuring robustness against overfitting to specific layouts or configurations.
> - **In-Domain Generalization:** While our VLM does not generalize to entirely unseen tasks, it exhibits strong generalization capabilities within the domain of trained tasks. This means that the VLM can effectively deal with variations of the same tasks, such as different layouts and arrangements of objects. The model’s ability to adapt to these variations within its training domain is a significant strength. It can understand and process tasks that are similar in nature to the trained ones but differ in specific details, thereby demonstrating a level of flexibility and adaptability within its operational domain.
>
> > Q3: How are the Stack Block Pyramid and Tower of Hanoi tasks randomized? As far as I can tell, MultiReAct will try to replicate exactly what has been done in the demonstration. If this is the case, would the task fail if the colors and initial ring locations are swapped?
> 1. **Stack Block Pyramid:**
>      In the Stack Block Pyramid task, randomization is introduced in terms of the block colors and starting positions. This approach ensures that while the overall task objective remains consistent, the specific scenario in each iteration is unique. MultiReAct is trained to understand the fundamental concept of building a pyramid, irrespective of these variations.
> 2. **Tower of Hanoi:**
> 	For the Tower of Hanoi task, we introduce randomization in the initial placement of the rings and occasionally in the color of the rings. The goal is to ensure that MultiReAct does not merely memorize a specific sequence of moves but understands the underlying logic and strategy required to solve the Tower of Hanoi under varying initial conditions.
>
> > Q4: Why do you use CLIP reward score as performance measure in section 4.3 but not the ground-truth reward?
>
> 1. **Ground-Truth Reward Challenges:**
> 	In the context of our experiments, defining a precise ground-truth reward can be challenging. The tasks we are addressing involve complex interactions within dynamic environments, making the establishment of a fixed, objective ground-truth reward for each possible outcome impractical. The subjective nature of some tasks further complicates this issue.
>  2. **CLIP Reward as a Proxy for Ground-Truth:**
> 	The CLIP reward score serves as a practical and effective proxy for ground-truth in our experiments. CLIP, trained on a vast range of images and text, is adept at evaluating the alignment between visual outcomes and textual descriptions. This alignment is crucial in tasks where success is defined by how well an action aligns with the given instructions or objectives.
>  3. **Real-World Applicability:**
> 	 Using the CLIP reward score aligns with real-world scenarios where ground-truth may not be readily available or easily quantifiable. By leveraging CLIP's capabilities, we simulate a more realistic setting where the system must evaluate its performance based on available visual and textual cues, mirroring practical applications.
>
>  If you find that our explanations and clarifications have resolved the issues you raised and have shed more light on the aspects you were unsure about, we kindly request you to consider revising your evaluation of our paper. We believe that our detailed responses might provide a clearer perspective on the value and robustness of our work.

---

> > ### Comment · Reviewer_ftHn · 2023-11-23
> >
> > I appreciate the authors’ effort in addressing my concerns in great detail.
> >
> > Upon reviewing the detailed experiment settings that the authors have included in this rebuttal, I agree that the comparisons to relevant baselines are justifiable, and that the designed experiment randomizations are sound.
> >
> > However, I’ll keep my original rating because all these experiment details should be made more transparent in the manuscript, and doing so might require a major revision.

---

### Meta-Review · Area_Chair_Wv9q · 2023-12-10

**Metareview:**

The authors propose an extension of the ReAct framework to encompass multimodal input, CLIP-powered reward signals, and low-level actors for embodied agent planning and acting. They evaluate on a vision-based robotics simulation environment and find a substantial benefit over various baselines.

Reviewers viewed the method as interesting and praised the motivation and that it unified existing, disparate frameworks such as CoT, ReAct, VLMs, and Cap. However there were major concerns about its presentation, as many experimental details were left out. Reviewers L9T3 and Ufhp also indicated that evaluation on a broader set of tasks is warranted, which I also agree with, given that one of the main promises of using pretrained vision-language models is to be able to generalize to a wide range of environments.

There was a large discrepancy in scores (3-8). When asked to comment, reviewers agreed that important experimental details had been left out of the paper, indicating that it may have been rushed. This is especially critical given the complex nature of their method (encompassing stitching together many pre-existing frameworks). It’s also disappointing that the authors never responded to reviewer Ufhp, who provided a detailed and well-thought out review. This ultimately reflected negatively on the paper, and in light of this as well as the above points, I don’t think it is ready for publication at ICLR. However I echo the reviewers’ comments that this is a great direction with promising results, and I hope to see future submissions incorporate the points that were brought up in this review cycle.

**Justification For Why Not Higher Score:**

This paper is not quite ready for publication, as it needs a few rounds of polishing to incorporate important experimental details as well as a broader set of environments.

**Justification For Why Not Lower Score:**

NA

---

### Decision · Program_Chairs · 2024-01-16

Reject